# ACTIVE LEARNING OVER MULTIPLE DOMAINS IN NATURAL LANGUAGE TASKS

## ABSTRACT

Studies of active learning traditionally assume the target and source data stem from a single domain. However, in realistic applications, practitioners often require active learning with multiple sources of out-of-distribution data, where it is unclear a priori which data sources will help or hurt the target domain. We survey a wide variety of techniques in active learning (AL), domain shift detection (DS), and multi-domain sampling to examine this challenging setting for question answering and sentiment analysis. We ask (1) what family of methods are effective for this task? And, (2) what properties of selected examples and domains achieve strong results? Among 18 acquisition functions from 4 families of methods, we find $\mathcal{H}$-Divergence methods, and particularly our proposed variant DAL-E, yield effective results, averaging 2-3% improvements over the random baseline. We also show the importance of a diverse allocation of domains, as well as room-for-improvement of existing methods on both domain and example selection. Our findings yield the first comprehensive analysis of both existing and novel methods for practitioners faced with multi-domain active learning for natural language tasks.

## 1 INTRODUCTION

New natural language problems, outside the watershed of core NLP, are often strictly limited by a dearth of labeled data. While unlabeled data is frequently available, it is not always from the same *source* as the *target* distribution. This is particularly prevalent for tasks characterized by (i) significant distribution shift over time, (ii) personalization for user subgroups, or (iii) different collection mediums (see examples in Section A).

A widely-used solution to this problem is to bootstrap a larger training set using active learning (AL): a method to decide which unlabeled training examples should be labeled on a fixed annotation budget (Cohn et al., 1996; Settles, 2012). However, most active learning literature in NLP assumes the unlabeled *source* data is drawn from the same distribution as the *target* data (Dor et al., 2020). This simplifying assumption avoids the frequent challenges faced by practitioners in *multi-domain active learning*. In this realistic setting, there are multiple sources of data (*i.e.* domains) to consider. In this case, it's unclear whether to optimize for homogeneity or heterogeneity of selected examples. Secondly, is it more effective to allocate an example budget per domain, or treat examples as a single unlabeled pool? Where active learning baselines traditionally select examples the model is least confident on (Settles, 2009), in this setting it could lead to distracting examples from very dissimilar distributions.

In this work we empirically examine four separate families of methods (uncertainty-based, $\mathcal{H}$-Divergence, reverse classification accuracy, and semantic similarity detection) over several question answering and sentiment analysis datasets, following (Lowell et al., 2019; Elsahar & Gallé, 2019b), to provide actionable insights to practitioners facing this challenging variant of active learning for natural language. We address the following questions:

1. What family of methods are effective for multi-domain active learning?
2. What properties of the example and domain selection yield strong results?

While previous work has investigated similar settings (Saha et al., 2011; Liu et al., 2015; Zhao et al., 2021; Kirsch et al., 2021) we contribute, to our knowledge, the first rigorous formalization and broad survey of methods within NLP. We find that many families of techniques for active learning and

domain shift detection fail to reliably beat random baselines in this challenging variant of active learning, but certain $\mathcal{H}$-Divergence methods are consistently strong. Our analysis identifies stark dissimilarities of these methods' example selection, and suggests domain diversity is an important factor in achieving strong results. These results may serve as a guide to practitioners facing this problem, suggesting particular methods that are generally effective and properties of strategies that increase performance.

## 2 RELATED WORK

**Active Learning in NLP** Lowell et al. (2019) shows how inconsistent active learning methods are in NLP, even under regular conditions. However, Dor et al. (2020); Siddhant & Lipton (2018) survey active learning methods in NLP and find notable gains over random baselines. Kouw & Loog (2019) survey domain adaptation without target labels, similar to our setting, but for non-language tasks. We reference more active learning techniques in Section 4.

**Domain Shift Detection** Elsahar & Gallé (2019b) attempt to predict accuracy drops due to domain shifts and Rabanser et al. (2018) surveys different domain shift detection methods. Arora et al. (2021) examine calibration and density estimation for textual OOD detection.

**Active Learning under Distribution Shift** A few previous works investigated active learning under distribution shifts, though mainly in image classification, with single source and target domains. Kirsch et al. (2021) finds that BALD, which is often considered the state of the art for unshifted domain settings, can get stuck on irrelevant source domain or junk data. Zhao et al. (2021) investigates *label shift*, proposing a combination of predicted class balanced subsampling and importance weighting. Saha et al. (2011), whose approach corrects joint distribution shift, relies on the *covariate shift assumption*. However, in practical settings, there may be general distributional shifts where neither the *covariate shift* nor *label shift* assumptions hold.

**Transfer Learning from Multiple Domains** Attempts to better understand how to handle shifted domains for better generalization or target performance has motivated work in question answering (Talmor & Berant, 2019; Fisch et al., 2019; Longpre et al., 2019; Kamath et al., 2020) and classification tasks (Ruder & Plank, 2018; Sheoran et al., 2020). Ruder & Plank (2017) show the benefits of both data similarity and diversity in transfer learning. Rücklé et al. (2020) find that sampling from a wide-variety of source domains (data scale) outperforms sampling similar domains in question answering. He et al. (2021) investigate a version of multi-domain active learning where models are trained and evaluated on examples from all domains, focusing on robustness across domains.

## 3 MULTI-DOMAIN ACTIVE LEARNING

Suppose we have multiple domains $D_1, D_2, ..., D_k$.[1] Let one of the $k$ domains be the **target** set $D_T$, and let the other $k - 1$ domains comprise the **source** set $D_S = \bigcup\limits_{i \neq T} D_i$.

**Given:**

- **Target:** Small samples of *labeled* data points $(x, y)$ from the **target** domain.
  $D_T^{train}, D_T^{dev}, D_T^{test} \sim D_T$.[2]
- **Source:** A large sample of *unlabeled* points $(x)$ from the **source** domains.
  $D_S = \bigcup\limits_{i \neq T} D_i$

**Task:**

1. **Choose** $n$ samples from $D_S$ to label.
   $D_S^{chosen} \subset D_S$, $|D_S^{chosen}| = n$, selected by $\arg\max_{x \in D_S} A_f(x)$ where $A_f$ is an acquisition function: a policy to select unlabeled examples from $D_S$ for labeling.
2. **Train** a model $M$ on $D^{final-train}$, validating on $D_T^{dev}$.
   $D^{final-train} = D_T^{train} \cup D_S^{chosen}$
3. **Evaluate** $M$ on $D_T^{test}$, giving score $s$.

---

[1]We define a domain as a dataset collected independently of the others.

[2]$|D_T^{train}| = 2000$ to simulate a small but reasonable quantity of labeled, in-domain training data for active learning scenarios.

For Step 1, the practitioner chooses $n$ samples with the highest scores according to their acquisition function $A_f$. $M$ is fine-tuned on these $n$ samples, then evaluated on $D_T^{test}$ to demonstrate $A_f$'s ability to choose relevant out-of-distribution training examples.

## 4 METHODS

We identify four families of methods relevant to active learning over multiple shifted domains. **Uncertainty methods** are common in standard active learning for measuring example uncertainty or familiarity to a model; $\mathcal{H}$-**Divergence** techniques train classifiers for domain shift detection; **Semantic Similarity Detection** finds data points similar to points in the target domain; and **Reverse Classification Accuracy** approximates the benefit of training on a dataset. A limitation of our work is we do not cover all method families, such as domain adaptation, just those we consider most applicable. We derive ∼18 active learning variants, comprising the most prevalent and effective from prior work, and novel extensions/variants of existing paradigms for the multi-domain active learning setting (see KNN, $\widetilde{RCA}$ and DAL-E).

Furthermore, we split the families into two acquisition strategies: **Single Pool Strategy** and **Domain Budget Allocation**. **Single Pool Strategy**, comprising the first three families of methods, treats all examples as coming from one single unlabeled pool. **Domain Budget Allocation**, consisting of **Reverse Classification Accuracy** methods, simply allocate an example budget for each domain.

We enumerate acquisition methods $A_f$ below. Each method produces a full ranking of examples in the source set $D_S$. To rank examples, most acquisition methods train an acquisition model, $M_A$, using the same model architecture as $M$. $M_A$ is trained on all samples from $D_T^{train}$, except for DAL and KNN, which split $D_T^{train}$ into two equal segments, one for training $M_A$ and one for an internal model. Some methods have both ascending and descending orders of these rankings (denoted by ↑ and ↓ respectively, in the method abbreviations), to test whether similar or distant examples are preferred in a multi-domain setting.

Certain methods use vector representations of candidate examples. We benchmark with both task-agnostic and task-specific encoders. The task-agnostic embeddings are taken from the last layer's CLS token in Reimers & Gurevych (2019)'s sentence encoder (Appendix for details). The task-specific embeddings are taken from the last layer's CLS token in the trained model $M_A$.

The motivation of the task-specific variant is that each example's representation will capture task-relevant differences between examples while ignoring irrelevant differences.[3] The versions of DAL and KNN methods that use task-specific vectors are denoted with "∗" in their abbreviation. Otherwise, they use task-agnostic vectors.

### 4.1 UNCERTAINTY METHODS

These methods measure the uncertainty of a trained model on a new example. Uncertainty can reflect either *aleatoric* uncertainty, due to ambiguity inherent in the example, or *epistemic* uncertainty, due to limitations of the model (Kendall & Gal, 2017). For the following methods, let $Y$ be the set of all possible labels produced from the model $M(x)$ and $l_y$ be the logit value for $y \in Y$.

**Confidence (CONF)**   A model's confidence $P(y|x)$ in its prediction $y$ estimates the difficulty or unfamiliarity of an example (Guo et al., 2017; Elsahar & Gallé, 2019a).

**Entropy (ENTR)**   Entropy applies Shannon entropy (Shannon, 1948) to the full distribution of class probabilities for each example, formalized as $A_{\text{ENTR}}$.

$$A_{\text{CONF}}(x, M_A) = -\max(P(y|x)) \qquad A_{\text{ENTR}}(x, M_A) = -\sum_{i=1}^{|Y|} P(y_i|x) \cdot \log P(y_i|x)$$

**Energy-based Out-of-Distribution Detection (ENG)**   Liu et al. (2020) use an *energy-based score* to distinguish between in- and out-distribution examples. They demonstrate this method is less susceptible to overconfidence issues of softmax approaches.

---

[3]For instance, consider in one domain every example is prefixed with "Text:" while the other is not — telling the difference is trivial, but the examples could be near-identical with respect to the task.

**Bayesian Active Learning by Disagreement (BALD)**  Gal & Ghahramani (2016) introduces estimating uncertainty by measuring prediction disagreement over multiple inference passes, each with a distinct dropout mask. BALD isolates *epistemic* uncertainty, as the model would theoretically produce stable predictions over inference passes given sufficient capacity. We conduct $T = 20$ forward passes on $x$. $\hat{y}_t = \text{argmax}_i P(y_i|x)_t$, representing the predicted class on the $t$-th model pass on $x$. Following (Lowell et al., 2019), ties are broken by taking the mean label entropy over all $T$ runs.

$$A_{ENG}(x, M_A) = -\log \sum_{y \in Y} e^{l_y} \qquad A_{\text{BALD}}(x, M_A) = 1 - \frac{\text{count}(\text{mode}_{t \in T}(\hat{y}_t))}{T}$$

## 4.2 $\mathcal{H}$-Divergence Methods

Ben-David et al. (2006; 2010) formalize the divergence between two domains as the $\mathcal{H}$-Divergence, which they approximate as the difficulty for a discriminator to differentiate between the two.[4] Discriminative Active Learning (DAL) applies this concept to the active learning setting (Gissin & Shalev-Shwartz, 2019).

We explore variants of DAL, using an XGBoost decision tree (Chen & Guestrin, 2016) as the discriminator model $g$.[5] For the following methods, let $D_T^{train-B}$ be the 1k examples from $D_T^{train}$ that were *not* used to train $M_A$. Let $E$ be an encoder function, which can be task-specific or agnostic as described above. We use samples both from $D_T^{train-B}$ and $D_S$ to train the discriminator. We assign samples origin labels $l$, which depend on the DAL variant. Samples from $D_S$ with discriminator predictions closest to 1 are selected for labeling. The acquisition scoring function for each DAL method and training set definition, respectively, are:

$$A_{\text{DAL}}(x, g, E) = g(E(x)) \qquad \{(E(x), l) \mid x \in D_T^{train-B} \cup D_S\}$$

**Discriminative Active Learning — Target (DAL-T)**  DAL-T trains a discriminator $g$ to distinguish between target examples in $D_T^{train-B}$ and out-of-distribution examples from $D_S$. For DAL-T, $l = \mathbb{1}_{D_T^{train-B}}(x)$.

**Discriminative Active Learning — Error (DAL-E)**  DAL-E is a novel variant of DAL. DAL-E's approach is to find examples that are similar to those in the target domain that $M_A$ misclassified. We partition $D_T^{train-B}$ further into erroneous samples $D_T^{err}$ and correct samples $D_T^{corr}$, where $D_T^{train-B} = D_T^{err} \cup D_T^{corr}$. For DAL-E, $l = \mathbb{1}_{D_T^{err}}(x)$.

## 4.3 Reverse Classification Accuracy

**RCA**  Reverse Classification Accuracy (RCA) estimates how effective source set $D_{i,i\in S}$ is as a training data for target test set $D_T$ (Fan & Davidson, 2006; Elsahar & Gallé, 2019b). Without gold labels for $D_i$ we compute soft labels instead, using the BERT-Base $M_A$ trained on the small labeled set $D_T^{train}$. We then train a child model $M_i$ on $D_i$ using these soft labels, and evaluate the child model on $D_T^{dev}$. RCA chooses examples randomly from whichever domain $i$ produced the highest score $s_i$.

$$A_{\text{RCA}} = \mathbb{1}_{D_{(\arg\max_{i \in S} s_i)}}(x) \qquad \widetilde{RCA}: \quad \tau_i = \frac{s_i}{s_T - s_i}, \; |D_i^{chosen}| = \frac{\tau_i}{\sum_j s_j}$$

**RCA-Smoothed ($\widetilde{RCA}$)**  Standard RCA only selects examples from one domain $D_i$. We develop a novel variant which samples from multiple domains, proportional to their relative performance on the target domain $D_T^{dev}$. RCA-smoothed ($\widetilde{RCA}$) selects $|D_i^{chosen}|$ examples from source domain $i$, based on the relative difference between the performance $s_i$ (of child model $M_i$ trained on domain $i$ with pseudo-labels from $M_A$) on the target domain, and the performance $s_T$ of a model trained directly on the target domain $D_T^{dev}$. Since these strategies directly estimates model performance on the target domain resulting from training on each source domain, RCA and $\widetilde{RCA}$ are strong **Domain Budget Allocation** candidates.

| MRQA Datasets | | | | | Sentiment Datasets | | | | |
|---|---|---|---|---|---|---|---|---|---|
| **Dataset** | **Q** | **C** | **\|Q\|** | **Q ⊥ C** | **Dataset** | **\|R\|** | **-** | **N** | **+** |
| SQuAD | Crowd | Wiki | 11 | ✗ | Amzn-Books | 144 | 12.1 | 8.8 | 79.1 |
| NewsQA | Crowd | News | 8 | ✓ | Amzn-Health | 80 | 9.3 | 7.0 | 83.7 |
| TriviaQA | Trivia | Web | 16 | ✓ | Amzn-Music | 132 | 36.2 | 9.1 | 54.7 |
| SearchQA | Jeopardy | Web | 17 | ✓ | Amzn-Software | 126 | 14.2 | 8.1 | 77.6 |
| HotpotQA | Crowd | Wiki | 22 | ✗ | Amzn-Sports | 84 | 49.9 | 0.0 | 50.1 |
| Natural-QS | Search | Wiki | 9 | ✓ | Amzn-Tools | 89 | 15.3 | 7.9 | 76.8 |
| | | | | | Imdb | 230 | 16.4 | 7.5 | 76.1 |
| | | | | | Yelp | 109 | 24.3 | 10.7 | 65.0 |

Table 1: **Datasets:** The question answering (left) and sentiment analysis (right) datasets in our experiments. Left: Query source (Q), Context source (C), mean query length ($|Q|$), and whether the query was written independently from the context ($Q \perp C$). Right: mean review length ($|R|$) and the percent representation of negative (-), neutral (N) and positive (+) labels.

### 4.4 NEAREST NEIGHBOUR / SEMANTIC SIMILARITY DETECTION (KNN)

Nearest neighbour methods (KNN) are used to find examples that are semantically similar. Using sentence encoders we can search the source set $D_S$ to select the top $k$ nearest examples by cosine similarity to the target set. We represent the target set as the mean embedding of $D_T^{train}$. For question answering, where an example contains two sentences (the query and context), we refer to KNN-Q where we only encode the query text, KNN-C where we only encode the context text, or KNN-QC where we encode both concatenated together. The acquisition scoring function per example, uses either a task-specific or task-agnostic encoder $E$:

$$A_{\text{KNN}}(x, E) = \text{CosSim}(E(x), \text{Mean}(E(D_T^{train})))$$

## 5 EXPERIMENTS

Experiments are conducted on two common NLP tasks: question answering (QA) and sentiment analysis (SA), each with several available domains.

**Question Answering** We employ 6 diverse QA datasets from the MRQA 2019 workshop (Fisch et al., 2019), shown in Table 1 (left).[6] We sample 60k examples from each dataset for training, 5k for validation, and 5k for testing. Questions and contexts are collected with varying procedures and sources, representing a wide diversity of datasets.

**Sentiment Analysis** For the sentiment analysis classification task, we follow (Blitzer et al., 2007) and (Ruder & Plank, 2018) by randomly selecting 6 Amazon multi-domain review datasets, as well as Yelp reviews (Asghar, 2016) and IMDB movie reviews datasets (Maas et al., 2011). [7] Altogether, these datasets exhibit wide diversity based on review length and topic (see Table 1). We normalize all datasets to have 5 sentiment classes: very negative, negative, neutral, positive, and very positive. We sample 50k examples for training, 5k for validation, and 5k for testing.

**Experimental Setup** To evaluate methods for the multi-domain active learning task, we conduct the experiment described in Section 3 for each acquisition method, rotating each domain as the target set. Model $M$, a BERT-Base model (Devlin et al., 2019), is chosen via hyperparameter grid search over learning rate, number of epochs, and gradient accumulation. The large volume of experiments entailed by this search space limits our capacity to benchmark performance variability due to isolated factors (the acquisition method, the target domain, or fine-tuning final models). However, our hyperparameter search closely mimics the process of an ML practitioner looking to select a best method and model, so we believe our experiment design captures a fair comparison among methods. See Algorithm 1 in Appendix Section B for full details.

---

[4]The approximation is also referred to as Proxy $\mathcal{A}$-Distance (PAD) from (Elsahar & Gallé, 2019b)

[5]Hyperparameter choices and training procedures are detailed in the Appendix.

[6]The workshop pre-processed all datasets into a similar format, for fully answerable, span-extraction QA: https://github.com/mrqa/MRQA-Shared-Task-2019.

[7]https://jmcauley.ucsd.edu/data/amazon/, https://www.yelp.com/dataset, https://ai.stanford.edu/~amaas/data/sentiment/.

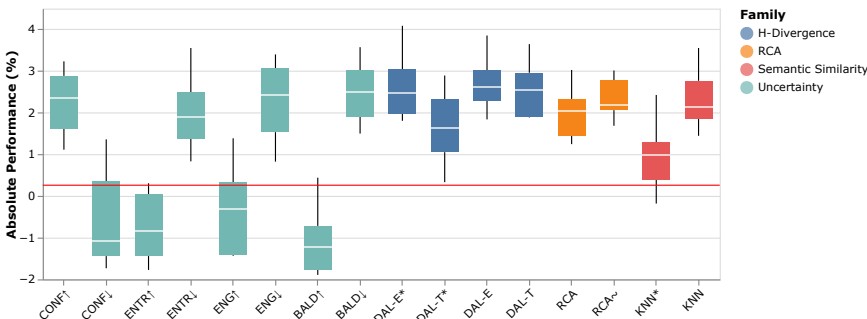

(a) **Sentiment Analysis** performance improvement (Accuracy %) by acquisition method.

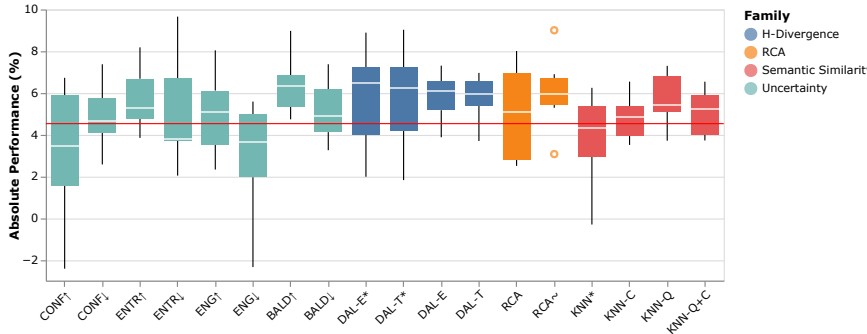

(b) **Question Answering** performance improvement (F1 %) by acquisition method.

Figure 1: **Performance by Method:** The improvement of each acquisition method over the model given no extra labelled data. Boxplot and whiskers denote the median, quartiles and min/max scores aggregated across each target domain and sample sizes ($n = \{8000, 18000, 28000\}$). The red line represents the median performance of a baseline that randomly selects examples to annotate.

# 6 RESULTS

## 6.1 COMPARING ACQUISITION METHODS

Results in Figure 1 show the experiments described in Section 5: benchmarking each acquisition method for multi-domain active learning. We observe for both question answering (QA) and sentiment analysis (SA), most methods manage to outperform the no-extra-labelled data baseline (0% at the y-axis) and very narrowly outperform the random selection baseline (red line). Consistent with prior work (Lowell et al., 2019), active learning strategies in NLP have brittle and inconsistent improvements over random selection. Our main empirical findings, described in this section, include:

- $\mathcal{H}$**-Divergence**, and particularly DAL-E variants, consistently outperform baselines and other families of methods.
- The ordering of examples in **Uncertainty methods** depend significantly on the diversity in source domains. BALD variants perform best among available options.
- Task-agnostic representations, used in ĸNN or **DAL** variants, provide consistently strong results on average, but task-specific representations significantly benefit certain target sets.
- Different families of methods rely on orthogonal notions of *relevance* in producing their example rankings.

$\mathcal{H}$**-Divergence** methods categorically achieved the highest and most reliable scores, both as a family and individual methods, represented in the top 3 individual methods 11 / 18 times for QA, and 20 / 24 times for SA. For QA, BALD↑ and DAL-E∗ had the best mean and median scores respectively, and for SA DAL-E achieved both the best mean and median scores. Among these methods, our proposed DAL-E variants routinely outperform DAL-T variants by a small margin on average, with equivalent training and tuning procedures. We believe this is because DAL-E captures both notions of domain similarity and uncertainty. By design it prioritizes examples that are similar to in-domain samples, but also avoids those which are uninformative, because the model already performs well on them.

Among **Uncertainty methods**, for SA methods which select for higher uncertainty vastly outperformed those which selected for low uncertainty. The opposite is true for QA. This suggests the

diversity of QA datasets contain more extreme (harmful) domain shift than the (mostly Amazon-based) SA datasets.[8] In both settings, the right ordering of examples with BALD (*epistemic* uncertainty) achieves the best results in this family of methods, over the others, which rely on *total* uncertainty.

Among **Reverse Classification Accuracy** methods, our $\widetilde{RCA}$ variant also noticeably outperforms standard RCA and most other methods, aside from DAL and BALD. Combining $\widetilde{RCA}$ with an example ranking method is a promising direction for future work, given the performance it achieves selecting examples randomly as a **Domain Budget Allocation** strategy.

Lastly, the **Semantic Similarity Detection** set of methods only rarely or narrowly exceed random selection. Intuitively, task-agnostic representations (κNN) outperform κNN∗, given the task-agnostic sentence encoder was optimized for cosine similarity.

**Embedding Ablations**    To see the effects of embedding space on κNN and DAL, we used both a task-specific and task-agnostic embedding space. While a task-specific embedding space reduces the examples to features relevant for the task, a task-agnostic embedding space produces generic notions of similarity, unbiased by the task model.

According to Figure 1, κNN outperforms κNN∗. In the QA setting, κNN∗'s median is below the random baseline's. In both plots, κNN∗'s whiskers extend below 0, indicating that in some cases the method actually chooses source examples that are harmful to target domain performance.

For DAL methods, task-agnostic and task-specific embeddings demonstrated mostly similar median performances. Notably, the boxes and whiskers are typically longer for task-specific methods than task-agnostic methods. This variability indicates certain target datasets may benefit significantly from task-specific embeddings, though task-agnostic embeddings achieve more consistent results.

**Comparing Example Rankings**    For each setting, we quantify how similar acquisition methods rank examples from $D_S$. In Figure 2, for each pair of methods, we calculate the Kendall's Tau coefficient between the source example rankings chosen for a target domain, then average this coefficient over the target domains. Kendall's Tau gives a scores $[-1, 1]$, with -1 meaning perfect anti-correlation, 0 meaning no correlation, and 1 meaning perfect correlation between the rankings. Methods from different families show close to no relationship, even if they achieve similar performances, suggesting each family relies on orthogonal notions of similarity to rank example relevance. This suggests there is potential for combining methods from different families for this task in future work.

In Sentiment tasks, all uncertainty methods had highly correlated examples. In QA, ENTR had little correlation with any method. This is likely due to the significantly larger output space for QA models. Compared to only 5 label classes in SA, question answering models distribute their start and end confidences over sequences of up to 512, where there can be multiple valid answer candidates. Embedding space also largely influences the examples that methods chose. DAL methods had higher correlations with each other when they share the same embedding space; *i.e.* DAL-E's ranking has a higher correlation with DAL-T than with DAL-E∗.

## 6.2 Properties of Optimal Example Selection

We examine three properties of optimally selected examples: (i) whether selecting from many diverse or one single domain leads to better performance, (ii) whether the selection of a domain or the individual examples matters more to performance, and (iii) whether selection strategies can benefit from source domain information rather than treating samples as drawn from a single pool? Our findings regarding properties of optimal selection, as described in this section, include:

- Selecting a diversity of domains usually outperforms selecting examples from a single domain.
- Acquisition functions such as DAL-E∗ do rely on example selection, mainly to avoid the possibility of large negative outcomes.
- **Domain Budget Allocation** during selection may improve performance. Surprisingly, even random selection from an "optimal" balance of domains beats our best performing acquisition methods most of the time.

---

[8] Accordingly, we attempt to derive a relationship between domain distance and method performance in Appendix F, but find intuitive calculations of domain distance uninterpretable.

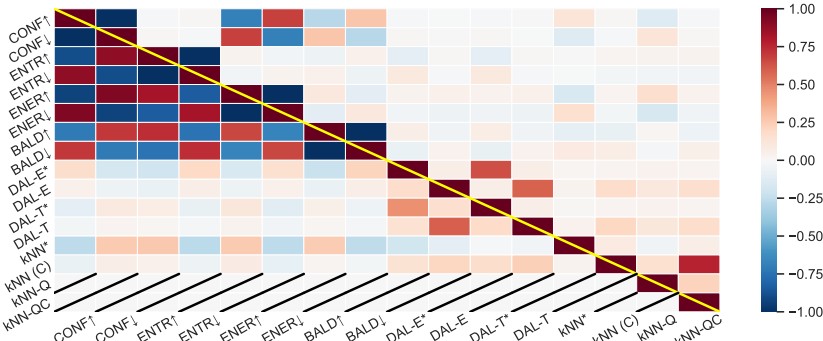

Figure 2: **Similarities of Example Rankings** Measured by Kendall's Tau Coefficients, for QA (above diagonal) and SA (below diagonal). Kendall's Tau coefficient is computed between the example rankings of each pair of methods. The heatmap contains these coefficients averaged over each target dataset (some cells are crossed out for SA since SA's κNN methods don't have C/Q/QC variants). 1 indicates a perfect relationship between the rankings, 0 means no relationship, and -1 means an inverse relationship.

**Are Many Diverse or One Single Domain Preferable?**   To answer this question we conduct a full search over all combinations of source datasets. For each target set, we fix 2k in-domain data points and sample all combinations of other source sets in 2k increments, such that altogether there are 10k training data points. For each combination of source sets, we conduct a simple grid search, randomly sampling the source set examples each time, and select the best model, mimicking standard practice among practitioners.

The result is a comprehensive search of all combinations of source sets (in 2k increments) up to 10k training points, so we can rank all combinations of domains per target, by performance. Tables 2a and 2b show the optimal selections, even as discrete as 2k increments, typically select at least two or more domains to achieve the best performance. However, 1 of 6 targets for QA, or 2 of 8 for the SA tasks achieve better results selecting all examples from a single domain, suggesting this is a strong baseline, if the right source domain is isolated. We also report the mean score of all permutations to demonstrate the importance of selecting the right set of domains over a random combination.

**Domains or Examples?**   Which is more important, to select the right domains or the right examples within some domain? From the above optimal search experiment we see selecting the right combination of domains regularly leads to strong improvements over a random combination of domains. Whether example selection is more important than domain selection may vary depending on the example variety within each domain. We narrow our focus to how much example selection plays a role for one of the stronger acquisition functions: DAL-E∗.

We fix the effect of domain selection (the number of examples from each domain) but vary which examples are specifically selected. Using DAL-E∗'s distribution of domains, we compare the mean performance of models trained on it's highest ranked examples against a random set of examples sampled from those same domains. We find a +0.46 ± 0.25% improvement for QA, and +0.12 ± 0.19% for SA. We also compare model performances trained on random selection against the lowest ranked examples by DAL-E∗. Interestingly, we see a -1.64 ± 0.37% performance decrease for QA, and -1.46 ± 0.56% decrease for sentiment tasks. These results suggest that example selection is an important factor beyond domain selection, especially for avoiding bad example selections.

**Single Pool or Domain Budget Allocation**   Does using information about examples' domains during selection lead to better results than treating all examples as coming from a single unlabeled pool? Originally, we hypothesized **Single Pool Strategy** methods would perform better on smaller budget sizes as they add the most informative data points regardless of domain. On the other hand, we thought that if the budget size is large, **Domain Budget Allocation** would perform best, as they choose source domains closest to the target domain. Based on Tables 1b and 1a, we were not able to draw conclusions about this hypothesis, as each sample size $n = \{8000, 18000, 28000\}$ produced roughly similar winning methods. Future work should include a wider range of budget sizes with larger changes in method performance between sizes.

| | Optimal Sample | | | | | | F1 Score | | | |
|---|---|---|---|---|---|---|---|---|---|---|
| | SQ | NE | TR | SE | HT | NQ | Optimal | Mean | Single Domain | Best AF |
| SQuAD | 2k | _8k_ | 0 | 0 | 0 | 0 | **78.0** | 74.0 | **78.0** | 77.0 |
| NewsQA | _6k_ | 2k | 0 | 2k | 0 | 0 | **56.1** | 52.0 | 55.2 | 55.9 |
| TriviaQA | _2k_ | 4k | 2k | 0 | 0 | 2k | **62.9** | 58.8 | 61.8 | 61.9 |
| SearchQA | 4k | 0 | 0 | 2k | _2k_ | 2k | 64.4 | 61.2 | 63.5 | **65.1** |
| HotpotQA | _6k_ | 0 | 0 | 0 | 2k | 2k | **67.1** | 63.6 | 66.4 | 66.0 |
| NaturalQ | _2k_ | 4k | 0 | 0 | 2k | 2k | 63.7 | 59.8 | 63.0 | **64.6** |
| MEAN | | | | | | | **65.4** | 61.5 | 64.6 | 65.1 |

(a) Optimal domain search for Question Answering (QA).

| | Optimal Sample | | | | | | | | Accuracy Score | | | |
|---|---|---|---|---|---|---|---|---|---|---|---|---|
| | A-B | A-H | A-M | A-So | A-Sp | A-T | IM | YE | Optimal | Mean | Single Domain | Best AF |
| AMZN-B | 2k | 0 | 0 | 2k | 2k | 0 | _4k_ | 0 | **69.0** | 66.5 | 67.6 | 68.7 |
| AMZN-H | 0 | 2k | 0 | 0 | 0 | _8k_ | 0 | 0 | 69.8 | 67.8 | 69.8 | **70.0** |
| AMZN-M | 0 | 0 | 2k | _0_ | 0 | 2k | 6k | 0 | **70.8** | 69.0 | 70.1 | 70.4 |
| AMZN-So | 2k | 0 | 2k | 2k | _4k_ | 0 | 0 | 0 | **64.7** | 62.6 | **64.7** | 64.4 |
| AMZN-Sp | 0 | 2k | 0 | 2k | 2k | _4k_ | 0 | 0 | 67.5 | 65.3 | 67.5 | **68.1** |
| AMZN-T | 0 | _8k_ | 0 | 0 | 0 | 2k | 0 | 0 | **68.4** | 65.7 | **68.4** | 68.3 |
| IMDB | _4k_ | 2k | 0 | 2k | 0 | 0 | 2k | 0 | 60.2 | 57.8 | 59.9 | **60.5** |
| YELP | _0_ | 2k | 0 | 4k | 2k | 0 | 0 | 2k | **67.0** | 64.9 | 66.1 | **67.0** |
| MEAN | | | | | | | | | **67.2** | 64.9 | 66.6 | **67.2** |

(b) Optimal domain search for Sentiment Analysis (SA).

Table 2: **Optimal Domain Search:** The optimal distribution of examples is shown per target domain, in 2k increments. The underlined value indicates the "Single source Domain" (2k in-domain, 8k source domain) that gave best results. On the right we show the F1 score for this *optimal* distribution, the *mean* score across all distribution combinations, the best *Single source Domain*, and the *Best Acquisition Function* (from Figure 1). Typically allocating *optimal* domain budgets and the *best acquisition functions* both performed strongly.

In our main set of experiments, the RCA acquisition functions follow the **Domain Budget Allocation** strategy, while all other acquisition functions follow the **Single Pool** strategies. Based on median performance, $\widetilde{RCA}$ outperformed all other methods (we're including BALD here due to inconsistency in performance between QA and SA) except for those in the $\mathcal{H}$-Divergence family. This suggests that using domain information during selection can lead to performance gains.

The Optimal Domain Search experiments, shown in Tables 2a and 2b, further suggest that allocating a budget from each domain can improve performance. For 8 out of our 14 experiments, selecting random samples according to the optimal domain distribution outperform any active learning strategy. While the optimal domain distributions were not computed a priori in our experiments, this result shows the potential for **Domain Budget Allocation** strategies. Future work could reasonably improve our results by developing an acquisition function that better predicts the optimal domain distributions than $\widetilde{RCA}$, or to even have greater performance gains by budgeting each domain, then applying an active learning strategy (e.g. DAL-E) within each budget.

## 7 CONCLUSION

We examine a challenging variant of active learning where target data is scarce, and multiple shifted domains operate as the source set of unlabeled data. For practitioners facing multi-domain active learning, we benchmark 18 acquisition functions, demonstrating the $\mathcal{H}$-Divergence family of methods and our proposed variant DAL-E achieve the best results. Our analysis shows the importance of example selection in existing methods, and also the surprising potential of domain budget allocation strategies. Combining families of methods, or trying domain adaptation techniques on top of selected example sets, offer promising directions for future work.

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

## A    MULTI-DOMAIN ACTIVE LEARNING TASK

In this section, we would enumerate real-world settings in which a practitioner would be interested in multi-domain active learning methods. We expect this active learning variant to be applicable to cold starts, rare classes, personalization, and settings where the modelers are constrained by privacy considerations, or a lack of labelers with domain expertise.

- In the **cold start** scenario, for a new NLP problem, there is often little to no target data available yet (labeled or unlabelled), but there are related sources of unlabelled data to try. Perhaps an engineer has collected small amounts of training data from an internal population. Because the data size is small, the engineer is considering out-of-domain samples, collected from user studies, repurposed from other projects, scraped from the web, etc..
- In the **rare class** scenario, take an example of a new platform/forum/social media company classifying hate speech against a certain minority group. Perhaps the prevalence of positive, in-domain samples on the social media platform is small, so an engineer uses out-domain samples from books, other social media platforms, or from combing the internet.
- In a **personalization** setting, like spam filtering or auto-completion on a keyboard, each user may only have a couple hundred of their own samples, but out-domain samples from other users may be available in greater quantities.
- In the **privacy constrained** setting, a company may collect data from internal users, user studies, and beta testers; however, a commitment to user privacy may incentivize the company to keep the amount of labeled data from the target user population low.
- Lastly, labeling in-domain data may require certain **domain knowledge**, which would lead to increased expenses and difficulty in finding annotators. As an example, take a text classification problem in a rare language. It may be easy to produce out-domain samples by labeling English text and machine translating it to the rare language, whereas generating in-domain labeled data would require annotators who are fluent in the rare language.

In each of these settings, target distribution data may not be amply available, but semi-similar unlabelled domains often are. This rules out many domain adaptation methods that rely heavily on unlabelled target data.

We were able to simulate the base conditions of this problem with sentiment analysis and question answering datasets, since they are rich in domain diversity. We believe these datasets are reasonable proxies to represent the base problem, and yield general-enough insights for a practitioner starting on this problem.

## B    REPRODUCIBILITY

### B.1    DATASETS AND MODEL TRAINING

We choose question answering and sentiment analysis tasks as they are core NLP tasks, somewhat representative of many classification and information-seeking problems. Multi-domain active learning is not limited to any subset of NLP tasks, so we believe these datasets are a reasonable proxie for the problem.

For question answering, the MRQA shared task (Fisch et al., 2019) includes SQuAD (Rajpurkar et al., 2016), NewsQA (Trischler et al., 2016), TriviaQA (Joshi et al., 2017), SearchQA (Dunn et al., 2017), HotpotQA (Yang et al., 2018), and Natural Questions (Kwiatkowski et al., 2019).

For the sentiment analysis classification task, we use Amazon datasets following (Blitzer et al., 2007) and (Ruder & Plank, 2018), as well as Yelp reviews (Asghar, 2016) and IMDB movie reviews datasets (Maas et al., 2011). [9]  Both question answering and sentiment analysis datasets are described in Table 1.

For reproducibility, we share our hyper-parameter selection in Table 3. Hyper-parameters are taken from Longpre et al. (2019) for training all Question Answering (QA) models since their parameters

---

[9]https://jmcauley.ucsd.edu/data/amazon/,    https://www.yelp.com/dataset, https://ai.stanford.edu/~amaas/data/sentiment/.

are tuned for the same datasets in the MRQA Shared Task. We found these choices to provide stable and strong results across all datasets. For sentiment analysis, we initially experimented on a small portion of the datasets to arrive at a strong set of base hyper-parameters to tune from.

Our BERT question answering modules build upon the standard PyTorch (Paszke et al., 2019) implementations from HuggingFace, and are trained on one NVIDIA Tesla V100 GPU.[10].

| Model Parameter | Value |
|---|---|
| Base Pre-trained Model | BERT-base |
| Model Size (# params) | $108.3M$ |
| Learning Rate | $5e-5$ |
| Optimizer | Adam |
| Gradient Accumulation | 1 |
| Dropout | 0.1 |
| Lower Case | False |
| **Question Answering model** | |
| Avg. Train Time | $2h\ 20m$ |
| Batch Size | 25 |
| Num Epochs | 2 |
| Max Query Length | 64 |
| Max Sequence Length | 512 |
| **Sentiment Classifcation model** | |
| Avg. Train Time | $43m$ |
| Batch Size | 20 |
| Num Epochs | 3 |
| Max Sequence Length | 128 |

Table 3: Hyperparameter selection for task models.

## B.2 EXPERIMENTAL DESIGN

For more detail regarding the experimental design we include Algorith 1, using notation described in the multi-domain active learning task definition.

---

**Algorithm 1** EXPERIMENTAL DESIGN

---

1: **for** each Acquisition Function $A_f$ **do**
2:    **for** each Target set $D_T \sim D$ **do**
3:       $D_T^{train}, D_T^{dev}, D_T^{test} \sim D_T$
4:       $D_S := \{x \in D \mid x \notin D_T\}$
5:       $M_A \leftarrow \text{TRAIN}(D_T^{train}, D_T^{dev})$
6:       $D^{chosen} \leftarrow [\text{Rank}_{x \in D_S} A_f(x, M_A)][: n]$
7:       $D^{final-train} = D_T^{train} \cup D^{chosen}$
8:       $M \leftarrow \text{GRIDSEARCH}(D^{final-train}, D_T^{dev})$
9:       $(A_f, D_T) = s_T^{A_f} \leftarrow M(D_T^{test})$
10:    **end for**
11: **end for**
12: **return** Scores Dictionary $(A_f, D_T) \to s_T^{A_f}$

---

## C ACQUISITION FUNCTIONS

### C.1 TASK AGNOSTIC EMBEDDINGS

To compute the semantic similarity between two examples, we computed the example embeddings using the pre-trained model from a sentence-transformer (Reimers & Gurevych, 2019). We used the

---

[10]https://github.com/huggingface/transformers

| Model Parameter DAL Discriminator | Value |
|---|---|
| Model Type | XGBoost |
| Model Size (# trees) | 10 |
| Model Size (maximum depth) | 2 |
| Learning Rate | 0.1 |
| Objective | binary:logistic |
| Booster | gbtree |
| Tree Method | gpu_hist |
| Gamma | 5 |
| Min Child Weight | 5 |
| Max Delta Step | 0 |
| Subsample | 1 |
| Colsample Bytree | 1 |
| Colsample Bynode | 1 |
| Reg Alpha | 0 |
| Reg Lambda | 5 |
| Scale Pos Weight | 1 |

Table 4: Hyperparameter selection for DAL discriminators.

RoBERTa large model, which has 24 layers, 1024 hidden layers, 16 heads, 355M parameters, and fine tuning on the SNLI (Bowman et al., 2015), MultiNLI (Williams et al., 2018), and STSBenchmark (Cer et al., 2017) datasets. Its training procedure is documented in `https://www.sbert.net/examples/training/sts/README.html`.

### C.2 BAYESIAN ACTIVE LEARNING BY DISAGREEMENT (BALD)

We note that Siddant and Lipton's presentation of BALD is more closely related to the Variation Ratios acquisition function described in Gal et al. (2017) than the description of dropout as a Bayesian approximation given in Gal & Ghahramani (2016). In particular, Gal et al. (2017) found that Variation Ratios performed on par or better than Houlsby's BALD on MNIST but was less suitable for ISIC2016.

### C.3 DISCRIMINATIVE ACTIVE LEARNING MODEL (DAL) TRAINING

DAL's training set is created using the methods detailed in Section 4.2. The training set is then partitioned into five equally sized folds. In order to predict on data that is not used to train the discriminator, we use 5-fold cross validation. The model is trained on four folds, balancing the positive and negative classes using sample weights. The classifier then predicts on the single held-out fold. This process is repeated five times so that each example is in the held out fold exactly once. Custom model parameters are shown in Table 4; model parameters not shown in the table are the default XGBClassifier parameters in xgboost 1.0.2. The motivations for choice in model and architecture are the small amount of target domain examples requiring a simple model to prevent overfitting and the ability of decision trees to capture collective interactions between features.

## D FULL METHOD PERFORMANCES

We provide a full breakdown of final method performances in Tables 5 and 6.

| Train Size | Target | random | CONF↑ | CONF↓ | ENTR↑ | ENTR↓ | ENG↑ | ENG↓ | BALD↑ | BALD↓ | DAL-E* | DAL-T* | DAL-E | DAL-T | RCA | $\widetilde{RCA}$ | KNN* | KNN-C | KNN-Q | KNN-QC |
|---|---|---|---|---|---|---|---|---|---|---|---|---|---|---|---|---|---|---|---|---|
| 10000 | HOTPOTQA | 65.76 | 64.15 | 64.38 | 64.59 | **66.03** | 65.39 | 62.39 | 63.13 | 61.45 | 65.42 | 65.33 | 63.58 | 63.18 | 65.19 | 65.33 | 62.25 | 64.28 | 63.98 | 63.51 |
| | NATURALQ | 63.05 | 61.59 | 62.14 | **64.61** | 63.56 | 61.52 | 62.44 | 58.35 | 61.56 | 63.0 | 62.79 | 62.54 | 62.7 | 58.72 | 62.73 | 59.75 | 61.94 | 63.28 | 61.84 |
| | NEWSQA | 53.51 | 47.72 | 51.82 | 54.14 | 52.85 | 52.54 | 48.06 | 55.61 | 52.76 | 51.36 | 50.93 | 54.41 | 54.69 | **55.93** | 54.31 | 50.77 | 53.13 | 55.52 | 52.91 |
| | SEARCHQA | 62.83 | 58.46 | 63.84 | 63.12 | 64.25 | 62.18 | 63.22 | 63.26 | **65.12** | 62.6 | 62.59 | 63.28 | 63.31 | 62.32 | 62.03 | 61.84 | 63.84 | 63.27 | 62.39 |
| | SQUAD | 75.97 | 73.23 | 75.33 | 76.28 | 73.41 | 76.22 | 73.07 | 75.65 | 73.13 | 76.61 | 76.75 | **77.0** | 76.88 | **77.0** | 76.25 | 76.74 | 74.24 | 75.08 | 74.94 |
| | TRIVIAQA | 61.44 | 58.19 | 60.17 | 59.75 | 57.57 | 59.64 | 59.4 | 60.02 | 58.32 | 61.89 | 61.24 | **61.94** | 61.06 | 58.88 | 60.81 | 60.45 | 59.98 | 60.82 | 60.37 |
| 20000 | HOTPOTQA | 66.29 | 64.12 | 64.3 | 65.15 | **67.53** | 65.86 | 63.86 | 67.51 | 63.76 | 67.05 | 67.13 | 64.48 | 64.23 | 65.81 | 66.78 | 61.96 | 64.14 | 64.68 | 64.13 |
| | NATURALQ | 63.62 | 63.65 | 62.12 | **64.87** | 64.11 | 63.3 | 60.32 | 64.86 | 63.63 | 63.99 | 64.14 | 63.98 | 63.38 | 59.21 | 63.76 | 60.34 | 61.54 | 63.81 | 62.43 |
| | NEWSQA | 54.71 | 48.32 | 52.68 | 55.44 | 54.78 | 53.01 | 47.56 | **57.69** | 55.2 | 52.15 | 52.1 | 55.62 | 56.29 | 57.33 | 57.47 | 50.16 | 53.55 | 55.5 | 54.94 |
| | SEARCHQA | 62.53 | 61.93 | 64.08 | 63.51 | 62.56 | 61.46 | 63.21 | 64.27 | **67.22** | 62.92 | 63.14 | 63.65 | 63.3 | 62.13 | 63.01 | 62.24 | 64.88 | 63.32 | 63.84 |
| | SQUAD | 76.32 | 75.33 | 75.53 | 77.61 | 72.17 | 76.54 | 72.79 | 78.02 | 74.15 | 77.51 | 77.72 | 77.7 | 77.59 | **78.57** | 78.0 | 77.79 | 75.93 | 76.56 | 76.27 |
| | TRIVIAQA | 62.45 | 61.37 | 61.97 | 61.64 | 61.21 | 62.38 | 60.2 | 61.74 | 60.54 | **63.38** | 62.56 | 62.7 | 61.99 | 59.76 | 61.65 | 62.54 | 61.83 | 62.08 | 62.84 |
| 30000 | HOTPOTQA | 65.98 | 64.79 | 66.33 | 64.43 | 68.3 | 65.76 | 63.39 | **69.17** | 63.44 | 67.09 | 67.51 | 64.91 | 65.34 | 65.92 | 67.79 | 62.32 | 64.09 | 65.85 | 64.68 |
| | NATURALQ | 63.61 | 63.49 | 63.18 | 64.51 | 64.65 | 63.87 | 62.68 | **66.4** | 63.62 | 64.66 | 65.12 | 64.84 | 64.24 | 59.18 | 63.64 | 61.63 | 62.32 | 64.24 | 62.66 |
| | NEWSQA | 55.18 | 47.73 | 54.26 | 56.79 | 54.48 | 54.62 | 48.38 | **58.4** | 56.7 | 53.48 | 53.48 | 55.63 | 56.17 | 57.7 | 56.84 | 49.19 | 54.89 | 56.24 | 54.54 |
| | SEARCHQA | 62.28 | 61.9 | 62.86 | 63.73 | 63.5 | 62.17 | 63.85 | 66.67 | **68.61** | 62.97 | 63.61 | 63.52 | 63.3 | 61.89 | 62.99 | 62.4 | 63.7 | 63.37 | 63.76 |
| | SQUAD | 77.75 | 74.1 | 76.78 | 76.98 | 75.08 | 76.76 | 73.08 | 78.7 | 77.04 | 79.21 | 78.71 | 78.08 | 79.24 | **80.18** | 78.38 | 78.76 | 75.77 | 77.88 | 77.13 |
| | TRIVIAQA | 63.2 | 62.34 | 61.98 | 62.01 | 61.87 | 62.98 | 60.13 | 61.85 | 62.49 | **64.36** | 64.35 | 63.21 | 62.97 | 61.36 | 62.81 | 63.22 | 62.94 | 62.91 | 63.89 |

Table 5: MQRA F1 scores from each active learning method over every training set size and target domain. The best performances are bolded and underlined.

| Train Size | Target | random | CONF↑ | CONF↓ | ENTR↑ | ENTR↓ | ENG↑ | ENG↓ | BALD↑ | BALD↓ | DAL-E* | DAL-T* | DAL-E | DAL-T | RCA | $\widetilde{RCA}$ | KNN* | KNN |
|---|---|---|---|---|---|---|---|---|---|---|---|---|---|---|---|---|---|---|
| 10000 | AMZN-B | 65.04 | **68.66** | 65.36 | 65.32 | 68.08 | 66.38 | 68.62 | 64.46 | 68.2 | 67.16 | 66.98 | 68.28 | 67.68 | 67.24 | 68.3 | 65.66 | 67.06 |
| | AMZN-H | 66.36 | 68.98 | 66.32 | 67.36 | 68.84 | 65.98 | 69.3 | 66.64 | 69.36 | **70.04** | 68.4 | 69.32 | 69.1 | 69.6 | 69.14 | 68.52 | 68.84 |
| | AMZN-M | 68.38 | 70.2 | 67.3 | 68.1 | 69.66 | 67.4 | 70.4 | 67.42 | 69.74 | **70.42** | 69.4 | 70.16 | 70.06 | 69.88 | 70.08 | 69.44 | 69.56 |
| | AMZN-SO | 61.06 | 63.94 | 61.92 | 61.38 | 64.32 | 62.42 | 64.3 | 61.24 | 64.3 | 63.46 | 63.04 | 64.2 | 64.22 | 62.4 | 64.12 | 63.72 | **64.42** |
| | AMZN-SP | 64.92 | 67.12 | 64.22 | 64.68 | 66.5 | 64.68 | 66.1 | 64.06 | 67.58 | 67.12 | 66.66 | 68.04 | 67.62 | **68.14** | 66.98 | 66.16 | 66.94 |
| | AMZN-T | 65.4 | 67.88 | 65.94 | 65.54 | 67.26 | 65.56 | 68.02 | 64.64 | 67.8 | **68.44** | 65.68 | 67.86 | 68.24 | 66.66 | 67.06 | 65.36 | 67.64 |
| | IMDB | 58.05 | 59.32 | 59.48 | 58.76 | 58.78 | 58.88 | 58.54 | 58.02 | 59.9 | 59.68 | 60.46 | 60.4 | 60.52 | 59.94 | 59.52 | 58.96 | 60.1 |
| | YELP | 66.75 | 64.94 | 63.82 | 64.36 | 65.58 | 63.46 | 65.88 | 64.42 | 66.38 | 66.06 | 66.4 | 65.84 | 66.98 | 66.0 | **67.04** | 66.24 | 65.46 |
| 20000 | AMZN-B | 64.68 | **69.12** | 65.92 | 65.18 | 68.46 | 67.08 | 69.04 | 66.5 | 68.88 | 68.18 | 65.64 | 68.16 | 68.68 | 67.9 | 67.88 | 66.26 | 67.64 |
| | AMZN-H | 67.16 | 69.46 | 65.04 | 64.94 | 69.54 | 65.94 | **70.32** | 65.32 | 69.84 | **70.32** | 68.08 | 69.94 | 70.04 | 70.16 | 70.28 | 67.66 | 69.18 |
| | AMZN-M | 68.76 | 70.86 | 66.2 | 67.84 | 69.98 | 66.18 | 70.82 | 66.7 | 70.52 | **71.48** | 69.56 | 70.84 | 70.54 | 71.32 | 69.86 | 68.78 | 70.28 |
| | AMZN-SO | 61.5 | 64.98 | 62.1 | 62.28 | **65.56** | 61.66 | 65.2 | 61.82 | 65.34 | 64.7 | 64.74 | 64.88 | 64.56 | 63.18 | 64.22 | 64.26 | 65.3 |
| | AMZN-SP | 65.68 | 67.18 | 65.04 | 63.78 | 67.14 | 65.66 | 66.34 | 63.52 | 67.36 | 68.22 | **68.78** | 68.36 | 68.72 | 68.42 | 67.54 | 65.32 | 67.84 |
| | AMZN-T | 65.92 | 68.1 | 66.26 | 65.04 | 68.44 | 65.52 | 68.18 | 64.76 | 69.02 | 69.62 | 65.76 | **69.72** | 69.1 | 67.12 | 68.38 | 66.6 | 68.58 |
| | IMDB | 58.58 | 59.56 | 58.88 | 58.38 | 58.74 | 59.74 | 58.76 | 58.84 | 58.96 | 60.2 | **60.76** | 60.1 | 60.06 | 59.94 | 60.54 | 59.38 | 59.98 |
| | YELP | 66.39 | 66.52 | 62.92 | 64.34 | 65.74 | 63.34 | 66.18 | 64.06 | 66.62 | **67.6** | 66.9 | 67.2 | 66.16 | 65.98 | 66.86 | 65.46 | 67.4 |
| 30000 | AMZN-B | 65.18 | 68.96 | 65.02 | 63.42 | 68.9 | 65.96 | 69.12 | 63.72 | **69.42** | 68.86 | 67.16 | 69.22 | 68.08 | 68.2 | 69.06 | 66.32 | 68.42 |
| | AMZN-H | 67.0 | **71.1** | 64.82 | 64.62 | 69.92 | 64.78 | 70.54 | 63.56 | 70.38 | 70.86 | 67.96 | 70.38 | 70.6 | 70.32 | 70.2 | 68.46 | 70.74 |
| | AMZN-M | 69.48 | 70.96 | 67.16 | 66.34 | 70.5 | 68.06 | 71.14 | 66.48 | 71.14 | **71.56** | 70.28 | 70.38 | 71.0 | 71.28 | 70.98 | 68.92 | 70.64 |
| | AMZN-SO | 62.94 | 66.06 | 62.0 | 61.56 | 66.06 | 61.76 | 65.98 | 60.52 | **66.36** | 65.88 | 66.0 | 65.58 | 65.8 | 63.44 | 65.98 | 64.58 | 66.22 |
| | AMZN-SP | 67.06 | 67.82 | 63.44 | 63.08 | 68.0 | 64.14 | 67.82 | 63.6 | 69.16 | 68.7 | 67.86 | **69.38** | 68.56 | 68.42 | 68.3 | 66.24 | 67.96 |
| | AMZN-T | 66.1 | 69.04 | 65.8 | 66.14 | 68.2 | 66.96 | 69.0 | 63.4 | 69.62 | **70.22** | 67.08 | 70.0 | 69.62 | 68.1 | 68.8 | 67.2 | 69.72 |
| | IMDB | 59.67 | 58.9 | 59.84 | 57.9 | 59.1 | 59.66 | 59.3 | 58.58 | 59.76 | 59.78 | **61.58** | 60.7 | 60.8 | 60.6 | 60.32 | 60.4 | 60.64 |
| | YELP | 66.93 | 66.28 | 63.68 | 64.28 | 66.7 | 63.38 | 67.34 | 63.62 | 67.16 | 66.78 | 67.46 | **68.2** | 66.94 | 66.22 | 67.18 | 65.54 | 67.82 |

Table 6: Sentiment accuracy scores from each active learning method over every training set size and target domain. The best performances are bolded and underlined.

# E KENDALL'S TAU

## E.1 DEFINITION

Kendall's Tau is a statistic that measures the rank correlation between two quantities. Let $X$ and $Y$ be random variables with $(x_1, y_1), (x_2, y_2), ..., (x_n, y_n)$ as observations drawn from the joint distribution. Given a pair $(x_i, y_i)$ and $(x_j, y_j)$, where $i \neq j$, we have:

$\frac{y_j - y_i}{x_j - x_i} > 0$ : pair is concordant

$\frac{y_j - y_i}{x_j - x_i} < 0$ : pair is discordant

$\frac{y_j - y_i}{x_j - x_i} = 0$ : pair is a tie

Let $n_c$ be the number of concordant pairs and $n_d$ the number of discordant pairs. Let ties add 0.5 to the concordant and discordant pair counts each. Then, Kendall's Tau is computed as:[11]

$$\tau = \frac{n_c - n_d}{n_c + n_d}$$

## E.2 INTER-FAMILY COMPARISON

Here, we extend on our comparison of example rankings by presenting plots of Kendall Tau scores normalized by intra-family scores in 3. For the sentiment setting, the ranges of intra-family Kendall Tau coefficients are smaller than the MRQA setting. Methods in the uncertainty family have especially strong correlations with each other and much weaker with methods outside of the family. For H-divergence based methods, intra-family correlations are not's as strong as for the uncertainty family;

---

[11] https://www.itl.nist.gov/div898/software/dataplot/refman2/auxillar/kendell.htm

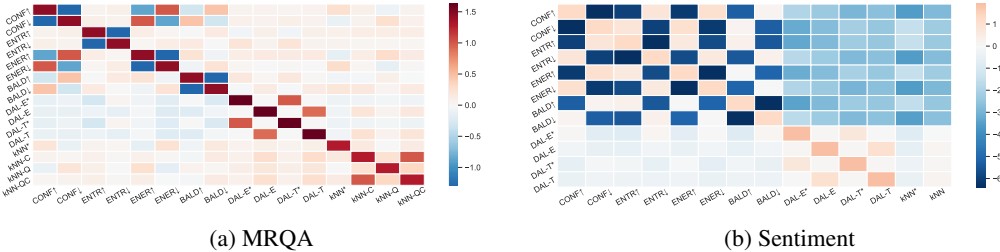

(a) MRQA                                    (b) Sentiment

Figure 3: Kendall Tau scores normalized by intra-family scores according to the family of the method on the y-axis (with uncertainty-ascending and uncertainty-descending as distinct families). If the cell's corresponding Kendall Tau score is within the intra-family range, it's value will be in $[0, 1]$. Below the range is negative, and above the range is greater than 1.

in fact, the Kendall Taus between DAL-E/κNN and DAL-T/κNN appear to be slightly within the H-divergence intra-family range.

Furthermore, intra-family ranges are quite large for all families in the MRQA setting. For each method, there is at least one other method from a different family with which it had a higher Kendall Tau coefficient than the least similar methods of its own family.

## F  RELATING DOMAIN DISTANCES TO PERFORMANCE

We investigated why certain methods work better than others. One hypothesis is that there exists a relationship between between target-source domain distances and method performance. We estimated the distance between two domains by computing the Wasserstein distance between random samples of 3k example embeddings from each domain. We experimented with two kinds of example embeddings: 1. A task agnostic embedding computed by the sentence transformer used in the κNN method, and 2. A task specific embedding computed by a model trained with the source domain used in the DAL∗ method. Given that there are $k - 1$ source domains for each target domain, we tried aggregating domain distances over its mean, minimum, maximum, and variance to see if Wasserstein domain distances could be indicative of relative performance across all methods.

Figure 4, Figure 5, Figure 6, and Figure 7 each show, for a subset of methods, the relationship between each domain distance aggregation and the final performance gap between the best performing method. Unfortunately, we found no consistent relationship for both MRQA and the sentiment classification tasks. We believe that this result arose either because our estimated domain distances were not reliable measures of domain relevance, or because the aggregated domain distances are not independently sufficient to discern relative performance differences across methods.

Figures 3-6: The average domain distance is calculated by finding the distance between 3k examples from $D_T$ and the combined set made from choosing 3k examples from each domain in $D_S$. Since the Wasserstein metric is symmetric, this yields $k$ points for comparison.

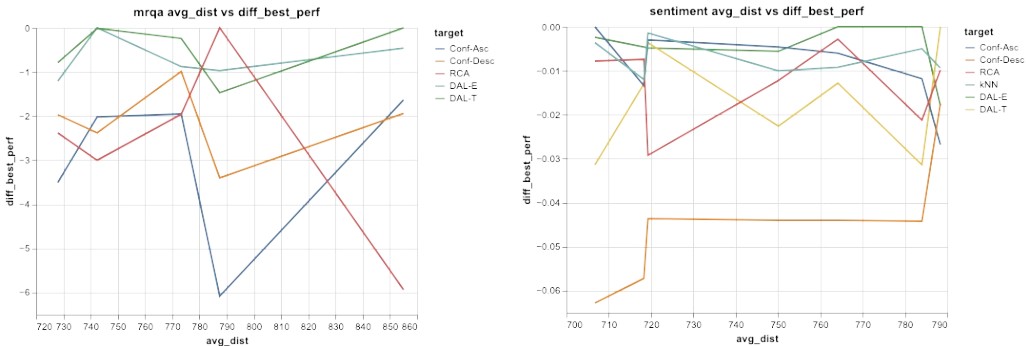

Figure 4: Average Wasserstein domain distance vs performance.

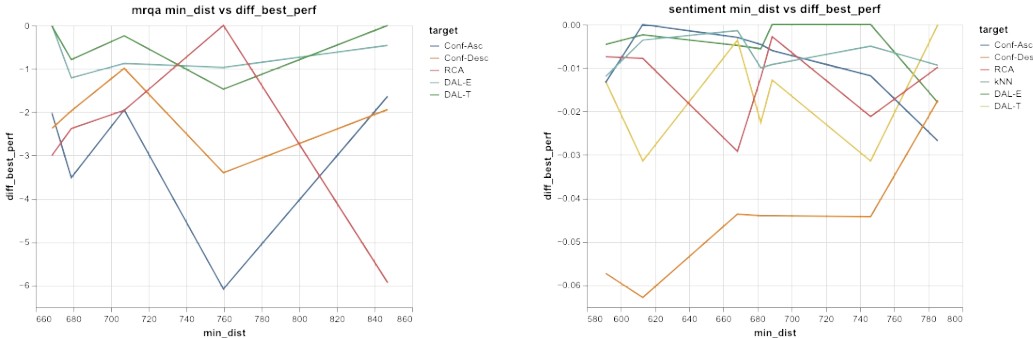

Figure 5: Minimum Wasserstein domain distance vs method performance.

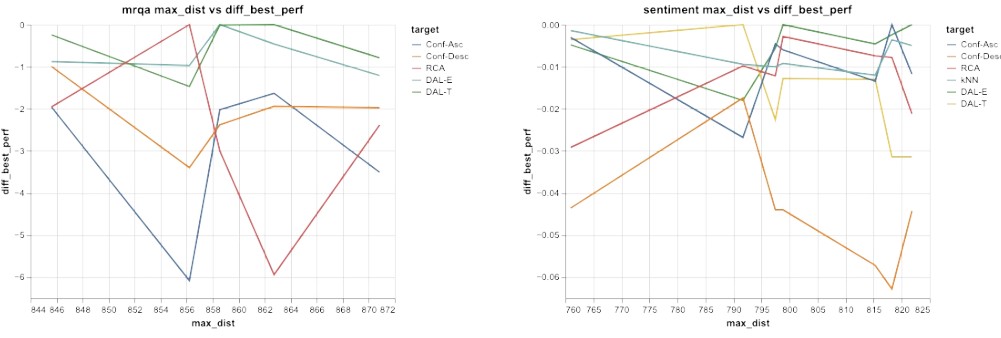

Figure 6: Maximum Wasserstein domain distance vs method performance.

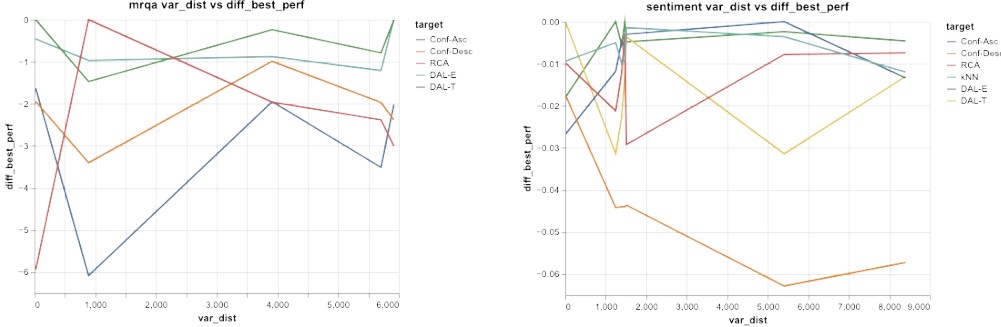

Figure 7: Wasserstein Domain distance variance vs performance.

