# OpenReview forum: "Active Learning over Multiple Domains in Natural Language Tasks"
_ICLR.cc/2022/Conference — ICLR 2022 Submitted_

### Official Review · Reviewer_EJMS · 2021-10-30

**Correctness:** 3
**Technical Novelty And Significance:** 2
**Empirical Novelty And Significance:** 2
**Recommendation:** 5
**Confidence:** 2

**Main Review:**

Reasons to accept:

- The study has a very ambitious goal. In the framework that the problem is defined, the authors, in my opinion, do a good job of defining sub-problems and validating the claims.

- The paper is framed and written well. The study inherently has many parts, consequently this makes following up the content a bit difficult. But it appears that the authors have recognized this challenge and structured the paper to minimize this issue for the reader, e.g., summarizing the findings at the beginning of each section.

- The paper is very detailed. Namely, the experiments are thorough, the discussions are extensive, and the references are comprehensive.

Reasons to reject:

- My first concern is about the topic itself. As a person who has experience in text classification. Has done research in Active Learning and also in Domain Adaptation. I am not clear in what real world situations one might need to do active learning with multiple-source domain adaptation. The authors cite some papers as the studies in this area. Some I already knew, and some I specifically checked. But to my knowledge none of them actually did research in this exact topic. Please note that the problems of label shift or dataset shift are different from what the authors are proposing here. At the beginning of the introduction section the authors provide a very short description. But I am still not clear about this. On the other hand, if this problem applies to only some very specific settings, then why not specifically write the paper for those settings. Because, currently the paper claims that the proposed problem is a general and common challenge.\
Another issue that further adds to the confusion is the data split used in the experiments. The authors assume each target domain already has several thousand labeled documents. So I can again ask, under what text classification scenarios there are several thousand labeled documents available but there is no unlabeled document available in that domain, so that we may be forced to use unlabeled documents from several other sources.

- My second concern is regarding some of the methods used in the experiments. I understand that preparing such a paper is difficult, but still to me it was not initially easy to relate the KNN methods and active learning. Or H-divergence methods and active learning. Of course, KNN relates to representativeness metric and H-divergence relates to data sampling. But the way that the authors frame these methods implies that these are commonly used in active learning methods. I disagree with the authors, the connection is not initially obvious at all. Another method used in the paper, which is still unclear to me, is to rank data points in an ascending order of uncertainty. It is still not clear to me why one should select the points with most confident label predictions. this contradicts the active learning purpose.

- My third concern is regarding the experimental setting: 1) The correct way of comparing active learning methods is to compare the learning curves. The authors only compare the final F1 measure. 2) None of the methods used in the paper are actual domain adaptation models. The authors simply aggregate the source and target data points to create a training set. This is not Domain Adaptation. 3) In H-divergence methods, authors use a decision tree-based classifier as discriminator. On the other hand the vectors are distributed representations, and the features are not independent. Tree-based classifiers assume feature independence.


Other less severe issues:

- Several questions that authors pose to empirically answer remained open and without answers, e.g., the questions in Section 6.2. There is nothing wrong with reporting failed expectations and failed methods. These are still informative. But when the number of these go up, it indicates that some of the choices were initially incorrect. This further confirms my concern about the lack of relation between some of the methods used in the paper and active learning.

- When the authors describe the method BALD, they cite a paper and state that the paper used dropout to quantify uncertainty. The paper that actually used dropout for this purpose is a different one [1].

- The two tasks used to evaluate the AL methods are very similar in nature.

- Not sure why the authors have used bold faced fonts so frequently. They also have used red font a few times, which is not necessary.


[1] Gal and Ghahramani. Dropout as a Bayesian approximation: Representing model uncertainty in deep learning. ICML 2016.

**Summary Of The Paper:**

The authors investigate the efficacy of several active learning-related techniques in classification under domain shift with multiple source domains. They include: uncertainty methods, H-divergence methods, reverse-classification methods, and nearest neighbor methods.

They construct 18 methods with various combinations and settings and carry out experiments in two text classification datasets: Question answering and sentiment analysis.

They report several findings, among them, the most interesting one to me is that uncertainty in various models manifests itself in various ways and candidate data points are not necessarily the same across models.

**Summary Of The Review:**

The paper is an exploratory study. It is well written and covers a wide range of experiments. But I question the application of the task, also in my opinion the connection between the used methods and active learning should be further discussed. Additionally, there are a few problems with the experimental setup.

---

> ### Author Response · Authors · 2021-11-15
> **Response to Reviewer EJMS**
>
> Thank you for your very generous review! We are glad the reviewer finds our goals ambitious, our experiments extensive, and our claims thoroughly validated. And we particularly appreciated the thoughtfulness of your critiques, which elicited much discussion regarding our framing. We address your central critiques below, and plan to include the results of this conversation directly in our next draft: (1) task setup, (2) choice of methods, and (3) experimental setup and relation to domain adaptation.
>
> ### Task Setup
>
> Thank you for raising this point. We expound on the motivation for our setup here, justifying the task for future readers. We will add this to the Introduction of the paper (with examples in the appendix for full clarity).
>
> First we should clarify: our evaluation sets are intentionally several thousand examples such that our final conclusions/inferences are statistically reliable [3]. However, we do not expect practitioners to mimic these splits, or have nearly as many eval examples available. (We will add this clarification to the paper.)
>
> For real-world settings, we expect multi-domain active learning to be applicable to cold starts, rare classes, personalization, and settings where the modelers are constrained by privacy considerations, or a lack of labelers with domain expertise.
>
> 1. In the cold start scenario, for a new NLP problem, there is often little to no target data available yet (labeled or unlabelled), but there are related sources of unlabelled data to try. Perhaps an engineer has collected small amounts of training data from an internal population. Because the data size is small, the engineer is considering out-of-domain samples, collected from user studies, repurposed from other projects, scraped from the web, etc.
> 2. In the rare class scenario, take an example of a new platform/forum/social media company classifying hate speech against a certain minority group. Perhaps the prevalence of positive, in-domain samples on the social media platform is small, so an engineer uses out-domain samples from books, other social media platforms, or from combing the internet.
> 3. In a personalization setting, like spam filtering or auto-completion on a keyboard, each user may only have a couple hundred of their own samples, but out-domain samples from other users may be available in greater quantities.
> 4. In the setting constrained by privacy, a company may collect data from internal users, user studies, and beta testers; however, a commitment to user privacy may incentivize the company to keep the amount of labeled data from the target user population low.
> 5. Lastly, labeling in domain-data may require certain domain knowledge, which would lead to increased expenses and difficulty in finding annotators. As an example, take a text classification problem in a rare language. It may be easy to produce out-domain samples by labeling English text and machine translating it to the rare language, whereas generating in-domain labeled data would require annotators who are fluent in the rare language.
>
> In each of these settings, target distribution data may not be amply available, but semi-similar unlabelled domains often are. We were able to simulate the base conditions of this problem with sentiment analysis and question answering, since they are rich in domain diversity. We believe the datasets we chose are reasonable proxies to represent the base problem, and yield general-enough insights for a practitioner starting on this problem. Given your feedback, we will be clear in the paper there are limitations to these results, given the wide breadth of possible settings in which multi-domain active learning may be applicable.

---

> > ### Author Response · Authors · 2021-11-15
> > **Response Part 2**
> >
> > ### Choice of Methods
> >
> > The reviewer asks why we include H-Divergence and high confidence methods in our experiments. The reason is because for multi-domain active learning, it isn’t clear a priori whether high uncertainty samples (most OOD) or low uncertainty samples (most ID) will benefit performance. So in addition to the traditional active learning methods sampling for high uncertainty (entropy, confidence, BALD), we include methods that intentionally sample low uncertainty or high similarity to the target, such as CONF-Descending (commonly used in domain shift detection [1] or selecting training examples from a heterogeneous pool [2]). Similarly, our H-Divergence methods build upon prior work in domain shift detection [1] (see PAD, which is roughly equivalent to DAL-T). Given your feedback, we will better motivate these techniques in the Methods section.
> >
> > ### Experimental Setup
> >
> > 1. Active Learning Curves -- We agree with the reviewer that many active learning papers qualitatively compare methods based on the learning curves. However, we believe our evaluation setup is equally as valid, and more indicative of a standard practitioner’s setup. Previous work samples tiny budgets (50-200 examples), label, retrain, repeat, but multiple rounds of labeling and training can be expensive, in terms of time and computation. We believe it’s more realistic for an engineer to launch a large labelling job (a few thousand samples), especially in a cold-start scenario.
> > 2. Domain Adaptation -- The reviewer raises a good point that we don’t incorporate  domain adaptation models into our setting; in the paper, our primary focus was evaluating active learning in the multi-domain setting. We list domain adaptation in related works as there are some overlapping concepts, including the use of a domain classifier in some cases. We clarify in bullet (1) we expect little labelled or unlabelled data in the target domain for practical multi-domain active learning setups. While we realize this does not rule out all potential domain adaptation solutions that might be attempted here, we believe these could be additive to the techniques we already explore. Practitioners will need to select a strong set of examples for the target domain, and then adapt their model using more sophisticated adaptation techniques. Nonetheless, we agree with the reviewer that it would be interesting to explore how domain adaptation methods could play a role in this setup, we hope to encourage this as a direction for future work.
> > 3. Decision Tree Classifiers -- We thought a decision tree classifier would be a natural fit for this problem as every explanatory variable in the tree is expected to interact with every variable further up the tree. To our knowledge, tree-based classifiers do not assume feature independence; every child node is predicated on the condition of the parent node. There are a couple prior works that suggest tree based classifiers (random forests) for domain shift detections: [4] and [5].
> >
> > Addressing your minor concerns:
> >
> > 1. Open Empirical Questions -- We absolutely acknowledge many questions remain open. This is a very challenging setting for active learning, and part of our goal is to demonstrate this very point: the difficulty and complexity of this task. We investigated many ambitious questions in this work, and prefer to be transparent and open about which ones were not easily resolved, leaving them for future work. We do hope the reviewers recognize the contribution of emphasizing where intuitive/traditional ML methods fail, as well as the several concrete findings we do converge on after significant examination of the problem.
> > 2. BALD Citation -- We thank the reviewer for catching this. We will update the citation.
> >
> > We thank the reviewer for their thoughtful and constructive feedback.
> >
> > References:
> >
> > 1. Hady Elsahar and Matthias Galle. “To annotate or not? predicting performance drop under domain shift.” (EMNLP 2019)
> > 2. David McClosky, Eugene Charniak, and Mark Johnson. “Effective self-training for parsing.” (NAACL 2006)
> > 3. Dallas Card, Peter Henderson, Urvashi Khandelwal, Robin Jia, Kyle Mahowald, and Dan Jurafsky. 2020. “With little power comes great responsibility.” (EMNLP 20202)
> > 4. Nair, Nimisha G., Pallavi Satpathy, and Jabez Christopher. "Covariate shift: A review and analysis on classifiers." 2019 Global Conference for Advancement in Technology (GCAT). IEEE, 2019.
> > 5. Xiong, Caiming, et al. "Latent domains modeling for visual domain adaptation." Proceedings of the AAAI Conference on Artificial Intelligence. Vol. 28. No. 1. 2014.

---

> > > ### Comment · Reviewer_EJMS · 2021-11-19
> > > **Re: Response Part 2**
> > >
> > > Thank you for the detailed response.
> > >
> > > I found some of the clarifications convincing—e.g., the applications of your study.
> > >
> > > In some other cases, you agreed to revise the paper—e.g., the selected tools and methods.
> > >
> > > However, I am not convinced with the justifications regarding the experiments:
> > >
> > > 1) It appears that the applications of your study mainly is in low-resource settings. Then, practically there is no justification to have thousands of labeled *training” data points. It makes sense to have a large *validation* set to properly *calibrate* the methods, so that methods are compared based on their maximum capacity. But having a large *training* set can affect the conclusions. Specifically in an active learning setting, where the methods are compared based on their learning trends, and not based on their final score.
> > >
> > > 2) Whether to use active learning curves or the final score. In my opinion, the final score is a useful piece of information, and it can be reported as a separate table in Appendix. But not as the central metric to compare methods. Historically, in CS venues, active learning methods are compared based on their learning curves, because various methods show different levels of efficacy at various amount of available training data. The final number is extremely biased, an does not tell us in what region we will cross the diminishing return point.
> > > I am hoping that the authors have stored the intermediate results, and for revising this part they do not have to re-run all of the experiments.
> > >
> > > 3) Regarding the missing domain adaptation methods. Most existing domain adaptation models heavily rely on the distribution of data. The sampling procedure in Active Learning forms this distribution in source domains. It is a very harsh simplification to leave Domain Adaptation off the experiments.
> > >
> > > 4) Regarding the decision tree. Thanks for correcting me. I used the word “dependence” in its literal meaning, and not in its probabilistic sense. I meant to say that the features of neural representations are individually meaningless, and they collectively form a meaningful space [1], that is why the representations are called “distributional representations”. Therefore, I am not sure how much fitting a decision tree on such a space, and making decisions based on individual features, makes sense.
> > >
> > > I will revise my evaluation to “marginally above the acceptance threshold”, if the authors agree to find a way to replace the final scores with the learning curves. The paper still makes harsh assumptions, but I am willing to do so in favor of the other merits of the paper.
> > >
> > > [1] Szegedy et al. Intriguing properties of neural networks. ICLR 2014.

---

> > > > ### Author Response · Authors · 2021-11-20
> > > > **Discussion of Reviewer EJMS Round 2 Feedback**
> > > >
> > > > Thank you for your quick response!
> > > >
> > > > ## Active Learning Evaluation
> > > >
> > > > We agree with the reviewer that there are good reasons to evaluate active learning methods based on learning curves, as seen in previous active learning papers. However, our goal was to keep the paper as close to what a practitioner would actually use active learning for. In our experience, repeatedly sampling small budgets (whether it's one sample to a couple thousand), labeling, retraining, and doing this for some number of iterations is an onerous process which isn't actually common in real world settings. It's more likely that a modeler is given a budget for labeling and goes through a single round of large scale annotation before retraining their model again -- especially for the cold-start scenario we described, pre-deployment.
> > > >
> > > > Unfortunately we do not have intermediate results in our evaluation because of the one-shot setup. However, to mirror our intended evaluation, we did examine three sampling budget levels: 10k, 20k, and 30k. While we openly acknowledge that including only 3 budget levels may be a limitation of the paper, we're hoping that having the 3 eases some of the reviewer's concerns about the final number being biased. In the main paper, we aggregate the results, but we could add plots that show results at each budget level if the reviewer thinks this will strengthen the paper. (Although when we did this initially, we did not see significant varying results -- and the distribution of examples selected showed little difference by Jenson-Shannon divergence).
> > > >
> > > > We again would like to thank the reviewer for their insightful feedback and conversation. Unfortunately we do not have the computational resources to re-run the massive set of experiments with more/iterative budget levels, to produce a learning curve, but please let us know if plots that more clearly display our 3 budget levels would help address this. If there's anything else we can do in the next few days to help you reconsider your evaluation, please let us know.
> > > >
> > > > ## Low-Resource Setting
> > > >
> > > > We still think that 2k in-domain training data is very reasonable for many realistic settings: such as collecting data internally over a few weeks from a small set of users (again fits the setting of small in-domain labelled data, without much in-domain unlabelled data). The reviewer is right that diminishing returns of data size is an important phenomenon to understand, but we’ve observed it is primarily task dependent: some tasks simply require more seed data than others to stabilize due to their difficulty. We believe this is a great avenue for more task-specific future work. Originally, we experimented with a couple sizes of seed dataset size (1k, 2k, 5k), for a subset of experiments, and settled on 2k as the most reasonable starting point, as 5k was starting to see diminishing returns for sentiment analysis, while we felt 1k may have been unreasonably low for a multi-domain task. We are happy to include a discussion of this limitation, incorporating the reviewers' thoughts.
> > > >
> > > > ## Domain Adaptation
> > > >
> > > > We agree it would be very interesting to see domain adaptation methods applied on top of the data selections, but unfortunately we do not have the computational resources to combine all our dataset permutations, parameter tuning and 18 methods with additional domain adaptation techniques. We do not see this as an assumption, but a reasonable limitation for one paper, which we will explicitly acknowledge in this draft and discuss as a great avenue for future work.
> > > >
> > > > ## Summary
> > > >
> > > > In summary, we really appreciate the depth of constructive feedback from these reviews. Accordingly, we are making significant revisions to improve our paper, including:
> > > > * Detailed justification of the real-world settings for multi-domain active learning, with several examples.
> > > > * Detailed justification of the selection of tasks and datasets as a reasonable proxy for this setting.
> > > > * Corrections to BALD citation (thank you!)
> > > > * Explicitly differentiate our work from domain adaptation, discussing limitations and future work in this direction.
> > > > * Justifying modeling choices more rigorously for H-Divergence/PAD methods.
> > > > * Justify evaluation setup, listing limitations, and adding observed outcomes with different seed training set sizes, and different sample budgets.
> > > >
> > > > Thank you!

---

### Official Review · Reviewer_cB6j · 2021-11-01

**Correctness:** 3
**Technical Novelty And Significance:** 2
**Empirical Novelty And Significance:** 3
**Recommendation:** 5
**Confidence:** 3

**Main Review:**

Overall, I like this work since it serves as a guide for practitioners faced with similar settings, and gives good guidelines on what works in these challenging domain shift settings. The experiments are  extensive and cover many of the domain shift settings in NLP. However, I think some of the analysis could be improved.

Weaknesses:
- I like the analysis on measuring how different methods rank examples. However, I would highly recommend performing some kind of normalization - e.g. to compare method a1 from family A and method b1 from family B, kendall’s tau score should be normalized w.r.t the score between methods from the same family.
- I would like to challenge some of the claims made in Section 6.2 of the paper. First, while it does seem like a diversity of examples helps, the boost is *very minimal* and it is certainly not true in general. Indeed, one can construct a domain distribution where getting data from several domains could hurt performance, if the mapping function (or the underlying P(y | x)) changes between domains.

Based on these weaknesses, I think the paper is currently borderline. However, if these are addressed i’m happy to adjust my scores.

**Summary Of The Paper:**

This paper surveys a broad range of techniques for active learning in the multi-domain setting applied to text - specifically, given a small labeled dataset from a target domain and a large amount of unlabeled data from a collection of source domains, how should examples be picked from the source domains to for labeling to maximise performance on the target domain. The main findings of the work are that selecting examples based on H-divergence measures perform much better than most active learning approaches. In-particular, they propose a method termed DAL-E where examples are selected from the source domain based on similarity to *misclassified* target domain examples, and show that it outperforms standard “disciminative active learning” on most of their experiments. From analysis, it is revealed that selecting from diverse domains helps, and their proposed DAL-E can help with avoiding bad examples to be selected.


**Summary Of The Review:**

The paper is well written and the experiments are useful for practitioners looking to apply active learning, however I think some of the analysis could be improved.

---

> ### Author Response · Authors · 2021-11-15
> **Response to Reviewer cB6j**
>
> Thank you for your positive comments and constructive feedback on our analysis! We appreciate you recognize our work as a beneficial guide for practitioners, with insights into what works in these challenging settings. We address your two concerns below, and offer the following revisions.
>
> ### Kendall Tau Normalization
>
> We were glad that you liked the analysis on different method example ranks. We think you have a great point about performing the normalization as this would more clearly show comparisons between inter and intra family example rankings. If we understand correctly, you are suggesting that we do something like:
> Let min(b) = minimum Kendall Tau coefficient between different methods in family b
> Let max(b) = maximum Kendall Tau coefficient between different methods in family b
> Let x = Kendall Tau score between a method from family a and a method from family b
> Then, we can set a normalized score to be z = (x - min(b)) / (max(b) - min(b))
> As a result, if the x < min(b), z will be negative. If x > max(b), z > 1. And if x is within the range of intra-family Kendall Tau scores, then z will be in [0, 1].
> If this isn’t what you’re suggesting, could you please clarify?
>
> We generated new plots with this normalization and here are the key insights we see:
> * For the sentiment setting, the ranges of intra-family Kendall Tau coefficients are smaller than the MRQA setting. Methods in the uncertainty family have especially strong correlations with each other and much weaker with methods outside of the family. For H-divergence based methods, intra-family correlations aren’t as strong as for the uncertainty family; in fact, the Kendall Taus between DALE/kNN and DALT/kNN appear to be slightly within the intra-family range.
> * Intra-family ranges are quite large for all families in the MRQA setting. In fact, for each method, there is at least one other method from a different family with which it had a higher Kendall Tau coefficient than the least similar methods of its own family.
>
> The additional insight on the difference between ranges of Kendall tau scores within families for the QA and sentiment settings extends our analysis in the paper. Our plan is to add the new plots to the appendix section, along with a paragraph containing these insights, and keep the original Kendall Tau plots in the main paper. This is our reasoning:
> * The normalized scores lose meaning about whether there is a positive, negative, or no correlation between example rankings, and we think this piece is important in the section.
> * Normalized score(method a, method b) != Normalized score(method b, method a). Additionally, we can’t normalize by the semantic similarity family in the sentiment setting since there are only 2 methods in this family and that would cause a denominator of 0 in the normalization equation. We’re concerned that the resulting heat map would be difficult to interpret (we’ll spend time thinking about whether there’s a better way to plot the results).
>
> ### Boost from Sample Diversity (6.2)
> The reviewer raises a good point. We mostly agree with this interpretation, and will update our discussion to emphasize that Single Domain sampling is a very robust baseline. A couple nuanced points we would also add to moderate that view slightly: firstly, the Single Domain baseline has a small search space so acts as a pretty precise oracle, whereas the “Optimal” domain selection is really a lower bound because of the coarse granularity of the (very computationally expensive) search space . This would suggest the margin between similar and diverse example selections are likely larger than what we empirically demonstrate. Secondly, domain diversity at training improves robustness to other domains as well, which is an orthogonal but pertinent point if the test set is small and there is uncertainty as to whether it is fully representative of the target domain.
>
> We will update the paper to reflect these discussions, if you agree! Thank you.

---

> > ### Author Response · Authors · 2021-11-23
> > **Last Response to Reviewer cB6j**
> >
> > On a final note, we have improved the paper by incorporating the changes requested by the reviewer:
> >
> > 1. Kendall-Tau normalized plots are now in the appendix with a much wider discussion (Section E). (We still plan to shuffle around parts of the discussion and plots between the main paper and appendix, but under limited time and space we wanted to get all the new additions written out in one section for the reviewer to see in the meantime.)
> > 2. We have edited the framing regarding sample diversity, the strength of the single domain baseline, and plan to incorporate more discussion around this.
> >
> > We thank the reviewer once again, and hope they find the changes to their satisfaction!
> > Best,

---

### Official Review · Reviewer_mKxw · 2021-11-02

**Correctness:** 3
**Technical Novelty And Significance:** 2
**Empirical Novelty And Significance:** 3
**Recommendation:** 6
**Confidence:** 4

**Main Review:**

Unlike most of previous work on active learning (AL) focusing on a single domain, this paper targets the multi-domain AL setting, a more realistic setting. The paper presents an extremely thorough study of how different active learning methods (18 acquisition functions from 4 different families of methods) and rigorous experiments & analysis (correlation analysis among rankings of examples, searching for the best combination of source domains) provides insights to several key questions regarding multi-domain AL.

Overall, the paper is very well written and I quite enjoyed reading it. However, with it being a great experimental report, my main concern is regarding the limited technical novelty of the work besides the two proposed simple variants (DAL-E derived from DAL-T, and RCA-smoothed derived from RCA, and all other are existing AL methods). Plus, part of the analysis on AL is also covered by previous work (e.g., Lowell et al., 2019).

Also in terms of providing a empirical study to multi-domain AL, it appears [1] also studies a similar problems and offers similar analysis. How much does the proposed work connect and differ from it? This is not discussed in the paper.

[1] Multi-Domain Active Learning: A Comparative Study. He et al. https://arxiv.org/pdf/2106.13516v1.pdf (June 2021)

**Summary Of The Paper:**

This paper studies active learning with multiple domains to select examples for two natural language tasks, sentiment analysis and question answering. By benchmarking 18 acquisition strategies from 4 families, the paper shows that overall the H-Divergence methods achieves the best improvements overall (including a proposed variant DAL-E, which considers both domain selection and example selection).

**Summary Of The Review:**

+ Good summary and thorough analysis of experiments studying 18 AL methods (2 of which are proposed variants) on 2 NLP tasks
+ Paper is well written

- Limited technical novelty (the two variants are simple modifications from existing methods)
- Similar analysis exists but not discussed

---

> ### Author Response · Authors · 2021-11-15
> **Response to Reviewer mKxw**
>
> Thank you for such generous feedback, and we are really glad you enjoyed reading our paper! We appreciate the reviewer found our study extremely thorough, unique from prior work, and “providing insights to several key questions regarding multi-domain AL”. We address your feedback below, and will make revision to the paper accordingly.
>
> 1 - Related work
>
> We thank the reviewer for directing us to the related work (He et al., 2021). This is indeed relevant work. The main and fundamental difference is they are evaluating generalization across all domains, not a specific target domain. And the pool of unlabelled examples they can choose from are the same as the target sets as well. This motivates a mostly different set of methods, and a primary focus on evaluating their multi-domain robustness. There are some interesting comparisons to be made in their observations that could be relevant for practitioners who face our variant of the setting, but are also concerned with robustness to more than one domain. We will cite and add this discussion in Introduction/Related Work.
>
> 2 - Technical Novelty
>
> Our primary contributions are definitely of empirical significance: emphasizing where intuitive/traditional active learning and ML methods fail, as well as analysis of what works and doesn’t. In terms of technical novelty, there are decades of work on active learning, so our hope was to build simple extensions of these methods for this task. Building on top of giants, we are pleased our methods, DAL-E and RCA-smoothed, remain simple but yield noticeable improvements. This technical simplicity is a big plus, particularly in active learning, where practitioners are likely to prefer easy and understandable implementations to quickly bootstrap a new labelling job.
>
> Thank you again, and let us know if these revisions adequately address your feedback!

---

> > ### Author Response · Authors · 2021-11-23
> > **Last Response to Reviewer mKxw**
> >
> > We thank the reviewer one last time for the very relevant citation, as well as their positive feedback.
> >
> > We have updated the paper accordingly. There are also several other significant additions (main paper and appendix) to address all reviewer's questions and comments. We hope you find the paper improved and appreciate your support!
> >
> > Best,

---

> > > ### Comment · Reviewer_mKxw · 2021-11-30
> > > **Thank you for the response**
> > >
> > > Thank you for the response. I appreciate the clarifications.

---

### Official Review · Reviewer_aTPg · 2021-11-03

**Correctness:** 3
**Technical Novelty And Significance:** 2
**Empirical Novelty And Significance:** 1
**Recommendation:** 3
**Confidence:** 2

**Main Review:**

This paper makes comparison with techniques used in active learning (AL), domain shift detection (DS), and multi-domain sampling to combine data from multiple sources. The experiments are conducted on datasets from questions answering and sentiment analysis.

The paper is well organized and easy to follow. However, the contribution of this paper is not clear. Specifically, I would expect authors provide more detailed recommendation for AL, DS, and multi-domain sampling in terms of sampling techniques, and population of different sources for certain application.

My another is concern is that the motivation of the experimental design is not clear. Why authors consider questions answering and sentiment analysis as the applications? It requires more analysis about experimental results, such as Figure 1 and tables in Section D.



**Summary Of The Paper:**

This paper makes comparison with techniques used in active learning (AL), domain shift detection (DS), and multi-domain sampling to combine data from multiple sources. The experiments are conducted on datasets from questions answering and sentiment analysis.

The paper is well organized and easy to follow. However, the contribution of this paper is not clear. Specifically, I would expect authors provide more detailed recommendation for AL, DS, and multi-domain sampling in terms of sampling techniques, and population of different sources for certain application.

My another is concern is that the motivation of the experimental design is not clear. Why authors consider questions answering and sentiment analysis as the applications? It requires more analysis about experimental results, such as Figure 1 and tables in Section D.



**Summary Of The Review:**

This paper makes comparison with techniques used in active learning (AL), domain shift detection (DS), and multi-domain sampling to combine data from multiple sources. The experiments are conducted on datasets from questions answering and sentiment analysis.

The paper is well organized and easy to follow. However, the contribution of this paper is not clear. Specifically, I would expect authors provide more detailed recommendation for AL, DS, and multi-domain sampling in terms of sampling techniques, and population of different sources for certain application.

My another is concern is that the motivation of the experimental design is not clear. Why authors consider questions answering and sentiment analysis as the applications? It requires more analysis about experimental results, such as Figure 1 and tables in Section D.

---

> ### Author Response · Authors · 2021-11-15
> **Response to Reviewer aTPg**
>
> Thank you for the review! We are encouraged that you found the paper organized and easy to follow, and we greatly appreciate the constructive feedback you provided. We address your main questions below, and provide several clarifications we can incorporate in our next revision.
>
> 1 - Significance of contributions
>
> Firstly, we hope the reviewer recognizes the formalization of this particular task variant as an important contribution, relevant to practitioners facing new NLP problems. And in terms of empirical analysis, we feel this work offers several significant contributions relevant to the details asked about for the reviewer: (1) A comprehensive comparison of existing and novel acquisition functions for active learning in multi-domain natural language settings, including their performances, consistency, and relative example selections; (2) New acquisition functions, including DAL-E, which yields more consistent and effective results compared to existing methods; (3) Results on the impact of domain allocation diversity, domain selection versus example selection, task relevance in embedding-dependent acquisition functions, and the relevance of domain information during example selection.
>
> For a challenging task, with little existing work, we feel these results cover a lot of existing questions, and set a solid foundation for future work. But given your feedback, we hope to convey that more clearly in our writing for future readers, and so are happy to adjust our framing/presentation if the reviewer has specific recommendations? We currently summarize the main points at the beginning of each Results section, and then discuss details thereafter.
>
> 2 - Sampling technique recommendations
>
> We would point the reviewer to section 6.1 where we compare and rank techniques in each method family, and across families. We recommend practitioners use DAL-E, or similar H-Divergence techniques, for their consistent empirical performance. BALD ranks second if the practitioner prefers Uncertainty-based Methods. We also make recommendations with respect to the type of sentence representation to use if the practitioner opts for H-Divergence or kNN techniques.
>
> 3 - Recommendations for populations of different sources
>
> To better answer the reviewer, we would like to clarify the question. If the reviewer is interested in comparing population sizes selected from different source domains by the methods, we illustrate and discuss this in Figure 2. If the reviewer is interested in optimal balances of domain selections, Tables 2a and b would be relevant. Alternatively, if the reviewer is curious of how the size of examples to select for labelling affects domain selection, we are happy to run that analysis and add that to the paper.
>
> 4 - Task choices
>
> We recognize that we did not specifically address our choice of natural language tasks and we thank the reviewer for bringing this to our attention. We choose question answering (QA) and sentiment analysis (SA) because they are core NLP tasks, with ample resources to simulate the multi-domain active learning setting. In the wild, this setting will often be encountered by practitioners facing new cold-start NLP problems, rare classes, personalization, or privacy constraints. It is impossible to choose a perfectly representative set of tasks because of the nature of these problems, but we believe QA and SA are reasonable proxies for an initial look into this problem. It is reasonable that many practitioners will face new NLP problems which are question answering-oriented or classification-based, and even if they aren’t, our empirical findings could offer a strong starting point for what methods to start with, and why. We will elaborate on these choices in the paper, and clearly state any limitations of these findings.
>
> Thank you again for your feedback! We would like to incorporate these proposed changes if you feel that they adequately answer your questions.

---

### Decision · Program_Chairs · 2022-01-20

**Decision:**

Reject

**Comment:**

This paper considers the problem of active learning (AL) with data drawn from multiple domains. This framing motivates integrating work on domain shift detection and adaptation into standard AL approaches.

The reviewers agreed that the work reports a robust set of experiments, which is a clear strength. However, they also raised key concerns, namely: (i) The heterogeneous setting considered is not particularly well motivated; (ii) The technical contributions of this work are limited. The latter would not be a major issue if the empirical evaluation addressed a clear open question (since this would constitute a useful contribution in and of itself), but the empirical contribution is somewhat limited given the unique setting considered and the relevant prior work (some of which seems to have been overlooked by the authors).